# Equivariant Adaptation of Large Pretrained Models

**Arnab Kumar Mondal**[*][†]
Mila, McGill University
ServiceNow Research

**Siba Smarak Panigrahi**[*]
Mila, McGill University

**Sékou-Oumar Kaba**
Mila, McGill University

**Sai Rajeswar**
ServiceNow Research

**Siamak Ravanbakhsh**
Mila, McGill University

## Abstract

Equivariant networks are specifically designed to ensure consistent behavior with respect to a set of input transformations, leading to higher sample efficiency and more accurate and robust predictions. However, redesigning each component of prevalent deep neural network architectures to achieve chosen equivariance is a difficult problem and can result in a computationally expensive network during both training and inference. A recently proposed alternative towards equivariance that removes the architectural constraints is to use a simple *canonicalization network* that transforms the input to a canonical form before feeding it to an unconstrained *prediction network*. We show here that this approach can effectively be used to make a large pretrained network equivariant. However, we observe that the produced canonical orientations can be misaligned with those of the training distribution, hindering performance. Using dataset-dependent priors to inform the canonicalization function, we are able to make large pretrained models equivariant while maintaining their performance. This significantly improves the robustness of these models to deterministic transformations of the data, such as rotations. We believe this equivariant adaptation of large pretrained models can help their domain-specific applications with known symmetry priors.

## 1 Introduction

Deep neural networks (DNNs) have demonstrated exceptional success in dealing with a range of data modalities, such as image and point cloud [1–5]. The majority of these applications require working with input data that undergoes various transformations, such as rotation, scaling, or translation in image data. Invariant and equivariant networks are "aware" of such transformations; they are specifically designed to ensure that the network's behaviour is insensitive to input transformations, leading to more accurate and robust predictions and improved sample complexity [e.g., 6, 7, 9–11, 13, 14]. As such, they have emerged as a promising solution to a wide range of computer vision and point cloud processing tasks [7, 9, 10, 15–20], including classification, object recognition, and segmentation, among others.

In parallel to this, it has been shown that scaling models, both in the number of parameters and amount of training data, systematically improve their performance [21, 23–28]. Pre-training on massive datasets has emerged as a practical method to harness this advantage. Several large models are now publicly available, and fine-tuning them on specific applications has proven to be a successful strategy in a variety of applications. However, such *foundation models* [29] are typically not equivariant

---

[*]These authors contributed equally to this work

[†]Correspondence: arnab.mondal@mila.quebec. Work done while at ServiceNow Research

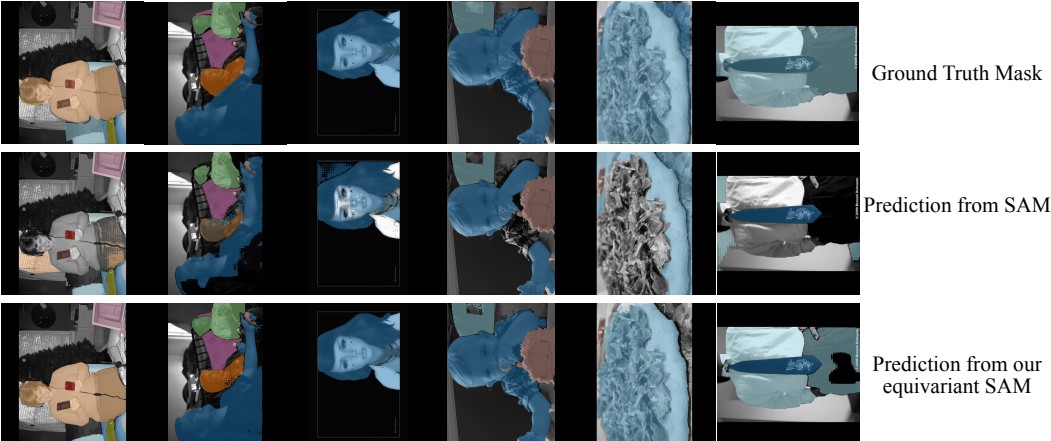

Ground Truth Mask

Prediction from SAM

Prediction from our
equivariant SAM

Canonicalization cost: $\Delta$ parameters = +0.3% ; $\Delta$ inference time: +7.3%

Figure 1: Predicted masks from the Segment Anything Model (SAM) [21], showcasing both the original model and our proposed equivariant adaptation for $90°$ counter-clockwise rotated input images taken from the COCO 2017 dataset [22]. Our method makes SAM equivariant to the group of $90°$ rotations while only requiring $0.3\%$ extra parameters and modestly increasing the inference time by $7.3\%$. For further insights into this experiment, refer to Section 4.1.1.

to most transformations except translations, as the currently known methods to achieve this are non-trivial to adapt to some architectures and remain computationally expensive.

This work aims at bridging the gap between foundation models and the systematic generalization offered by equivariance. The key idea is to decouple equivariance from the main task-specific DNN. One way of achieving this is through the idea of learned *canonicalization* function [30], where a canonicalization network learns to transform the data into a canonical form. The pretrained prediction network is then applied to this canonical form. Another way to achieve this decoupling is through the idea of *symmetrization* where all the transformations of a datapoint are passed through the prediction network and the final output is computed as the average of these predictions [12, 8]. However, symmetrization comes at a notably higher computational cost compared to canonicalization, primarily due to the requirement of a forward pass through the prediction network for each data transformation. This problem is even more pronounced when employing a large-scale prediction network. Hence, the canonicalization technique is a more efficient and practical choice for making large pretrained network equivariant.

Nevertheless, a naive application of the canonicalization idea fails in practice. This is because the canonicalization network's choice of canonical form can result in a change of distribution in the input of the pretrained prediction network – that is, canonicalization can be uninformed about the preference of the prediction network, undermining its performance.

As outlined in Figure 2, we resolve this misalignment by matching the distribution of predicted canonical forms with the dataset. We demonstrate empirically that imposing this prior is essential in an equivariant adaptation of pretrained models across different domains and datasets. Our approach offers a practical solution to obtaining large-scale equivariant models by providing an independent module that can be plugged into existing large pretrained DNNs, making them equivariant to a wide range of transformation groups.

## 2   Background

**Groups and Transformations**   Groups capture the abstract structure of symmetry transformations, which is due to their compositional form – e.g., the composition of a transformation with its inverse results in an identity transformation. Formally, a group $\mathcal{G}$ is a set equipped with an associative binary operation, such that the set is closed under this operation, and each element $g \in \mathcal{G}$ has a unique inverse. A group $\mathcal{G}$ can *act* on a set $\mathcal{X}$ by transforming its elements $\mathbf{x} \in \mathcal{X}$ through a bijection. Assuming $\mathcal{X}$

is a vector space, the linear action of $g$ on $\mathbf{x}$ is given by $\rho(g) \cdot \mathbf{x}$. The function $\rho : \mathcal{G} \rightarrow \mathrm{GL}(\mathcal{X})$ is called a group representation, and $\mathrm{GL}(\mathcal{X})$ is the set of invertible linear transformations on $\mathcal{X}$. For example, if $\mathcal{G} = SO(2)$ is the group of 2D rotations, its action on any image $\mathbf{x} \in \mathcal{X}$ could rotate it around some center.

**Equivariance in Deep Learning** In the context of deep learning, we are interested in designing models that are invariant or equivariant to the input data's symmetries. Suppose that input data live in a space $\mathcal{X}$ with some symmetry structure that we would like to preserve in our deep learning model.

An equivariant function $f : \mathcal{X} \rightarrow \mathcal{Y}$ is a function that commutes with the group action of $G$ on $\mathcal{X}$ and $\mathcal{Y}$, i.e., for all $g \in G$ and $\mathbf{x} \in \mathcal{X}$, we have:

$$f(\rho(g) \cdot \mathbf{x}) = \rho'(g) \cdot f(\mathbf{x}) \tag{1}$$

where $\rho' : G \rightarrow \mathcal{T}'$ is another group representation that acts on the output space $\mathcal{Y}$. In other words, an equivariant function is a function whose output changes in a predictable way under transformations of the input induced by the group $\mathcal{G}$. If $\rho'$ is the trivial representation, i.e., $\rho'(g) = I$ or identity transformation for all $g \in \mathcal{G}$, then we say that $f$ is invariant to the group action of $\mathcal{G}$.

**Equivariance with Learned Canonicalization** The concept of invariance involves ensuring that all elements within a group orbit produce the same output when processed by a function $f$. To achieve this, elements can be mapped to a canonical sample from their orbit prior to applying the function. On the other hand, equivariance involves mapping elements to a canonical sample, applying the function, and then transforming them back to their original position within the orbit. This notion can be precisely expressed by representing the equivariant function $f$ in a canonicalized form as follows:

$$f\left(\mathbf{x}\right) = \rho'(c\left(\mathbf{x}\right)) \cdot p\left(\rho(c\left(\mathbf{x}\right)^{-1}) \cdot \mathbf{x}\right) \tag{2}$$

where the function $p : \mathcal{X} \rightarrow \mathcal{Y}$ is called the *prediction function* and the function $c : \mathcal{X} \rightarrow \mathcal{G}$ is called the *canonicalization function*. In simpler terms, the function $c$ maps input data to group elements such that the inverse action of these elements can return the data back to its canonical sample.

The function $f$ in eq. (2) achieves equivariance for any prediction function as long as the canonicalization function is also $\mathcal{G}$-equivariant – that is $c\left(\rho\left(g\right) \cdot \mathbf{x}\right) = gc\left(\mathbf{x}\right) \ \forall \ g, \mathbf{x} \in \mathcal{G} \times \mathcal{X}$. Previous work by [30] used a direct approach to design a canonicalization function using existing equivariant neural network architectures.

This formulation offers the advantage of removing the constraint on the main prediction network and placing it instead on the network that learns the canonicalization function. As shown in [30], this decoupling can be partial, where the prediction network is itself equivariant to some symmetry transformations, and the canonicalization function is used for equivariance to additional symmetries.

## 3   Method

The flexibility of the equivariance with canonicalization approach enables the conversion of any existing large pretrained Deep Neural Network (DNN) into an equivariant model with respect to certain known transformations. To achieve this, the pretrained DNN can be utilized as the prediction network in the formulation provided by eq. (2). Subsequently, a canonicalization function can be designed to produce the elements of the known group of transformation while maintaining equivariance with respect to this group.

One could learn the canonicalization function while optionally finetuning the prediction function using the same task objective – in our experiments, we consider both zero-shot and fine-tuning setup.

The performance of the model requires the *alignment* of these two networks. For example, if the canonicalization network produces images that are upside-down, compared to those that the pretrained network is expecting, the overall performance is significantly degraded. In addition to alignment, there is an *augmentation* effect that further muddies the water: during its training, the canonicalization network performs data augmentation. As we see shortly, one needs to consider both alignment and augmentation effects when analyzing the performance of this type of equivariant network.

When both networks are trained together from scratch, the alignment is a non-issue, and (unwanted) augmentation can degrade or improve performance, depending on the extent of symmetry in the

dataset. However, when dealing with pretrained prediction networks one needs to also consider the alignment effect. One could then think of freezing the pretrained prediction network, therefore avoiding unwanted augmentation, and backpropagating the task loss through it to align the canonicalization network. However, this can become prohibitively expensive for large pretrained models, such as segment anything (SAM) considered in this work. We propose an alternative, where we directly regularize the canonicalization network to produce canonical forms consistent with the training data, which in turn aligns with the prediction network.

## 3.1 Learning Canonicalization, Augmentation and Alignment

When learning the canonicalization function during training, the process implicitly performs dynamic augmentation of the prediction network. Consider a model designed to be equivariant to a certain group of transformations $\mathcal{G}$ by canonicalization. At the start of training, the randomly initialized weights of the canonicalization function will output random canonical orientations for each data point. This has a similar effect to data augmentation using the group $\mathcal{G}$ for the prediction network. As training progresses, the canonical orientations for similar-looking images begin to converge, as demonstrated in Figure 3 in Appendix C, causing the augmentation effect to diminish. Thus, in addition to guaranteeing equivariance, this formulation provides a free augmentation effect to the prediction network.

Table 1: Effect of augmentation on the Prediction network. Top-1 classification accuracy and $\mathcal{G}$-Averaged classification accuracy for CIFAR10 and CIFAR100 [31]. $C8$-Avg Acc refers to the top-1 accuracy on the augmented test set obtained using the group $\mathcal{G} = C8$, with each element of $\mathcal{G}$ applied on the original test set.

| Dataset → | | CIFAR10 | | CIFAR100 | |
|---|---|---|---|---|---|
| Prediction Network ↓ | Rotation Augmentation | Acc | $C8$-Avg Acc | Acc | $C8$-Avg Acc |
| ResNet50 [32] | $-10$ to $+10$ degrees | $\mathbf{90.96 \pm 0.41}$ | $44.87 \pm 0.60$ | $\mathbf{74.83 \pm 0.15}$ | $37.14 \pm 0.42$ |
| | $-180$ to $+180$ degrees | $84.60 \pm 1.83$ | $81.04 \pm 1.86$ | $61.07 \pm 0.27$ | $59.42 \pm 0.70$ |
| | Learned Canonicalization (LC) [30] | $83.11 \pm 0.35$ | $\mathbf{82.89 \pm 0.41}$ | $59.84 \pm 0.67$ | $\mathbf{59.45 \pm 0.49}$ |
| ResNet50 [32] (pretrained on ImageNet) | $-10$ to $+10$ degrees | $\mathbf{96.97 \pm 0.01}$ | $57.77 \pm 0.25$ | $\mathbf{85.84 \pm 0.10}$ | $44.86 \pm 0.12$ |
| | $-180$ to $+180$ degrees | $94.91 \pm 0.07$ | $90.11 \pm 0.19$ | $80.21 \pm 0.09$ | $74.12 \pm 0.05$ |
| | Learned Canonicalization (LC) [30] | $93.29 \pm 0.01$ | $\mathbf{92.96 \pm 0.09}$ | $78.50 \pm 0.15$ | $\mathbf{77.52 \pm 0.07}$ |

However, there are two scenarios where the augmentation provided by learning the canonicalization function can be detrimental:

First, in cases where the training only requires small transformations as augmentations, providing all the transformations of a group $\mathcal{G}$ can actually hurt the performance of the prediction network during the start of *training* or *fine-tuning*. For example, in natural image datasets like CIFAR10, small rotation augmentations (from $-10$ to $+10$ degrees) are beneficial, while a canonicalization function would output any rotation from $-180$ to $+180$ degrees during the beginning phase of training. This can lead to unstable training of the prediction network and hinder the model's performance by training on additional data that is far outside the distribution of the train set. We show this in Table 1 by training a prediction network with different rotation augmentations, including the one due to learned canonicalization on both CIFAR datasets. Furthermore, we also observe that this effect is more pronounced when the prediction network is trained from scratch, and the dataset is more complicated with a larger number of classes. This effect can be understood as an example of the *variance-invariance tradeoff* introduced by [33]. Since, the test distribution is not perfectly symmetric under rotations, training with arbitrary augmentations biases the prediction function.

Second, we notice that relying solely on the task loss objective is not be sufficient for the canonicalization function to learn the correct orientation. This could be due to the small size of the finetuning dataset compared to the pretraining dataset. We see experimentatlly that this leads to the canonicalization function outputting inconsistent canonical orientations during inference, impacting the prediction network's performance. For instance, on CIFAR10 (non-augmented), we expect the canonical orientation for every datapoint to be similar after training. However, from Figure 4 in Appendix C, we can see that the canonical orientations for the test set are distributed uniformly from $-180$ to $+180$ degrees even after training until convergence of the task objective. As a result, during inference,

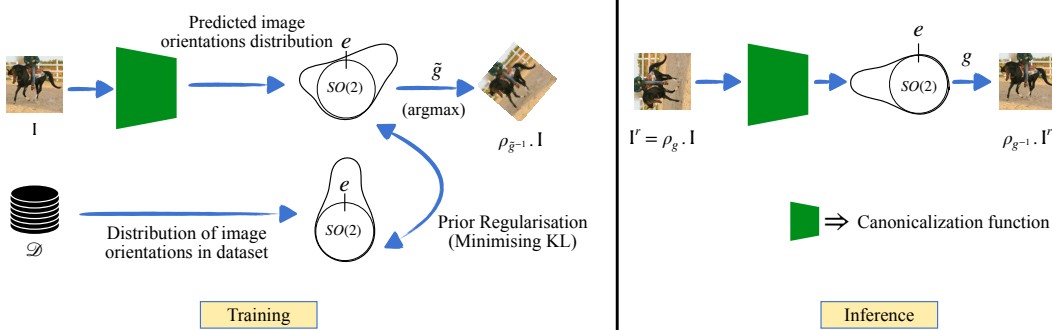

Figure 2: Training and inference with our proposed regularized canonicalization method. The canonicalization function outputs a distribution over image orientations used to canonicalize the input image. Additionally, during training, this predicted distribution is regularized to match the orientations seen in the dataset.

the prediction network will view images with different orientations and underperform. This issue arises from the fact that the prediction networks are not inherently robust to these transformations.

### 3.2 Canonicalization Prior

As we have seen, the canonicalization network may induce a shift in canonical orientations away from those present in the pretraining datasets. To further encourage the canonicalization function to align inputs in an orientation that will help the prediction network, we introduce a simple regularizer we call canonicalization prior (CP). It is motivated by noticing that the images or point clouds in the finetuning dataset provide useful information, as we expect these orientations to be similar to those of the training dataset. We, therefore, take as prior that the canonicalization should align inputs as closely as possible to their original orientation in the finetuning dataset.

We derive the regularizer by taking a probabilistic point of view. The canonicalization function maps each data point to a distribution over the group of transformations, denoted as $\mathcal{G}$. Let $\mathbb{P}_{c(\mathbf{x})}$ denote the distribution induced by the canonicalization function over $\mathcal{G}$ for a given data point $\mathbf{x}$. We assume the existence of a canonicalization prior associated with the dataset $\mathcal{D}$ that has a distribution $\mathbb{P}_{\mathcal{D}}$ over $\mathcal{G}$. To enforce the canonicalization prior, we seek to minimize the Kullback-Leibler (KL) divergence between $\mathbb{P}_{\mathcal{D}}$ and $\mathbb{P}_{c(\mathbf{x})}$ over the entire dataset $\mathcal{D}$ that is $\mathcal{L}_{\text{prior}} = \mathbb{E}_{\mathbf{x} \sim \mathcal{D}} \left[ D_{KL}(\mathbb{P}_{\mathcal{D}} \parallel \mathbb{P}_{c(\mathbf{x})}) \right]$.

We assume that the canonicalization function $c$ estimates the parameters of a distribution over rotations with probability density function $p(\mathbf{R} \mid c(\mathbf{x}))$. We denote the probability density function of the prior to be $q(\mathbf{R})$. Since the prior distribution is kept constant, minimizing the KL divergence is equivalent to minimizing the cross-entropy, and the prior loss simplifies to:

$$\mathcal{L}_{\text{prior}} = -\mathbb{E}_{\mathbf{x} \sim \mathcal{D}} \, \mathbb{E}_{\mathbf{R} \sim q} \left[ \log p(\mathbf{R} \mid c(\mathbf{x})) \right] \tag{3}$$

Hereafter, we derive the regularization for the discrete and continuous cases separately.

### 3.2.1 Discrete Rotations

We first consider the group of 2D discrete rotations, the cyclic group $C_n$, which can be seen as a discrete approximation to the full rotation group $SO(2)$. In this case, we consider a categorical distribution over group elements, with the prior distribution having probability mass of 1 for the identity element. Then $p_{\mathcal{D}}(\mathbf{R}) = \delta_{\mathbf{R},\mathbf{I}}$, where $\delta_{\mathbf{R},\mathbf{I}}$ is the Kronecker delta function and the cross entropy in eq. (3) becomes $-\log p_{c(x)}(\mathbf{I})$. Hence, the loss becomes $\mathcal{L}_{\text{prior}} = -\mathbb{E}_{x \sim \mathcal{D}} \log p_{c(x)}(\mathbf{I})$. In other words, the regularization loss is simply the negative logarithm of the probability assigned by the canonicalization function to the identity element $\mathbf{I}$ of the group.

**Practical Implementation** For images, similar to [30], the canonicalization network needs to output logits corresponding to every group element in the discrete cyclic group $C_n$. This can be achieved by using a Group Convolutional Network [7] or an $E(2)$-Steerable Network [34] that produces outputs using *regular* representation. To design the canonicalization function, we take a spatial average and

get logits corresponding to every element in the group along the fibre dimension. This is similar to the approach used in [30]. Now, we can get a discrete distribution over the group elements by taking a softmax and minimizing the prior loss along with the task objective. During training, to ensure consistency with the implementation in [30] for fair comparisons across all experiments, we utilize the argmax operation instead of sampling from this distribution using Gumbel Softmax [35] and employ the straight through gradient trick [36]. All our image-based experiments use this discrete rotation model.

### 3.2.2 Continuous rotations

When considering canonicalization with continuous rotations, it is natural to use the matrix Fisher distribution introduced by [37]. It is the analogue of the Gaussian distribution on the $SO(n)$ manifold and is defined as

$$p\left(\mathbf{R} \mid \mathbf{F}\right) = \frac{1}{n\left(\mathbf{F}\right)} \exp\left(\operatorname{Tr}\left[\mathbf{F}^T \mathbf{R}\right]\right) \tag{4}$$

where $\mathbf{F} \in \mathbb{R}^{n \times n}$ is the parameter of the distribution and $n\left(\mathbf{F}\right)$ is a normalization constant. Interpretation of the parameter $\mathbf{F}$ and useful properties of the distribution are provided in [38–40]. In particular, considering the proper singular value decomposition $\mathbf{F} = \mathbf{U}\mathbf{S}\mathbf{V}^T$, we find that $\hat{\mathbf{R}} \equiv \mathbf{U}\mathbf{V}^T$ is the mode of the distribution and the singular values $\mathbf{S}$ can be interpreted as concentration parameters in the different axes. We therefore set $\mathbf{S} = s\mathbf{I}$ to obtain the isotropic version of the distribution,

$$p\left(\mathbf{R} \mid \hat{\mathbf{R}}, s\right) = \frac{1}{n\left(s\right)} \exp\left(s \operatorname{Tr}\left[\hat{\mathbf{R}}^T \mathbf{R}\right]\right) \tag{5}$$

where the normalization constant only depends on $s$ (Theorem 2.1 of [39]). Note that on $SO\left(2\right)$, this becomes the Von-Mises distribution as expected.

We introduce the following result, which will allow to derive the regularization.

**Proposition 1.** *Let $p$ and $q$ be matrix Fisher distributions of $\mathbf{R}$*

$$p\left(\mathbf{R} \mid \hat{\mathbf{R}}_p, s_p\right) = \frac{1}{n\left(s_p\right)} \exp\left(s_p \operatorname{Tr}\left[\hat{\mathbf{R}}_p^T \mathbf{R}\right]\right), \quad q\left(\mathbf{R} \mid \hat{\mathbf{R}}_q, s_q\right) = \frac{1}{n\left(s_q\right)} \exp\left(s_q \operatorname{Tr}\left[\hat{\mathbf{R}}_q^T \mathbf{R}\right]\right).$$

*The cross-entropy is given by*

$$\mathbb{E}_{\mathbf{R} \sim q}\left[\log p\left(\mathbf{R} \mid \hat{\mathbf{R}}_p, s_p\right)\right] = N\left(s_q\right) s_p \operatorname{Tr}\left(\hat{\mathbf{R}}_p^T \hat{\mathbf{R}}_q\right) + \log c\left(s_p\right) \tag{6}$$

*where $N\left(s_q\right)$ only depends on $s_q$.*

The proof follows in Appendix A

Setting the location parameters of the estimated and prior distributions as $\mathbf{R}_{c(x)}$ and $\hat{\mathbf{R}}_q = \mathbf{I}$ respectively, we find that the canonicalization prior eq. (3) is given by

$$\mathcal{L}_{\text{prior}} = -\lambda \operatorname{Tr}\left(\mathbf{R}_{c(x)}\right) = \frac{\lambda}{2} \left\|\mathbf{R}_{c(x)} - \mathbf{I}\right\|_F \tag{7}$$

where we have eliminated terms that do not depend on $\mathbf{R}_{c(x)}$ and $\lambda = N\left(s_q\right) s_p$. Following intuition, the strength of the regularization is determined by the concentrations of the distributions around their mode.

**Practical Implementation**   For image domain, canonicalization network needs to output rotation matrices $\mathbf{R}_{c(x)} \in SO(2)$ that equivariantly transforms with the input image. This can be achieved by using a $E(2)$-Steerable Network [34] that outputs two vector fields. To design the canonicalization function we can take a spatial average over both vector fields and Gram Schmidt orthonormalize the vectors to get a 2D rotation matrix. While this sounds promising in theory, in practice we found it empirically difficult to optimize using the loss to enforce canonicalization prior (see Appendix D). We

Table 2: Performance comparison of large pretrained models finetuned on different vision datasets. Both classification accuracy and $\mathcal{G}$-averaged classification accuracies are reported. Acc refers to the accuracy on the original test set, and $C8$-Avg Acc refers to the accuracy on the augmented test set obtained using the group $\mathcal{G} = C8$. $C8$-Aug. refers to fine-tuning the pre-trained model with rotation augmentations restricted to $C8$.

| Pretrained Large Prediction Network → | | ResNet50 | | ViT | |
|---|---|---|---|---|---|
| Datasets ↓ | Model | Acc | $C8$-Avg Acc | Acc | $C8$-Avg Acc |
| CIFAR10 [31] | Vanilla | **96.97 ± 0.01** | 57.77 ± 0.25 | **98.13 ± 0.04** | 63.59 ± 0.48 |
| | Rotation Augmentation | 94.91 ± 0.07 | 90.11 ± 0.19 | 96.26 ± 0.15 | 93.67 ± 0.39 |
| | Learned Canonicalization (LC) [30] | 93.29 ± 0.01 | 92.96 ± 0.09 | 95.00 ± 0.01 | 94.80 ± 0.02 |
| | $C8$-Aug. | 95.76 ± 0.07 | 94.36 ± 0.09 | 96.36 ± 0.02 | 94.17 ± 0.08 |
| | Prior-Regularized LC (ours) | 96.19 ± 0.01 | **95.31 ± 0.17** | 96.14 ± 0.14 | **95.08 ± 0.10** |
| CIFAR100 [31] | Vanilla | **85.84 ± 0.10** | 44.86 ± 0.12 | **87.91 ± 0.28** | 55.87 ± 0.14 |
| | Rotation Augmentation | 80.21 ± 0.09 | 74.12 ± 0.05 | 82.59 ± 0.44 | 78.39 ± 0.89 |
| | Learned Canonicalization (LC) [30] | 78.50 ± 0.15 | 77.52 ± 0.07 | 80.86 ± 0.17 | 80.48 ± 0.20 |
| | $C8$-Aug. | 83.00 ± 0.09 | 79.72 ± 0.10 | 83.45 ± 0.09 | 80.08 ± 0.38 |
| | Prior-Regularized LC (ours) | 83.44 ± 0.02 | **82.09 ± 0.09** | 84.27 ± 0.10 | **83.61 ± 0.01** |
| STL10 [42] | Vanilla | **98.30 ± 0.01** | 73.87 ± 1.43 | **98.31 ± 0.09** | 76.66 ± 0.93 |
| | Rotation Augmentation | 98.08 ± 0.06 | 94.97 ± 0.08 | 97.85 ± 0.17 | 94.07 ± 0.11 |
| | Learned Canonicalization (LC) [30] | 95.30 ± 0.19 | 93.92 ± 0.10 | 95.11 ± 0.01 | 94.67 ± 0.02 |
| | $C8$-Aug. | **98.31 ± 0.01** | 96.31 ± 0.13 | 97.83 ± 0.08 | 94.45 ± 0.35 |
| | Prior-Regularized LC (ours) | 97.01 ± 0.01 | **96.37 ± 0.12** | 96.15 ± 0.05 | **95.73 ± 0.16** |

believe this warrants further investigation and present a potential novel research direction to explore. However, in the domain of point clouds, we discovered that combining the implementation from [30], which outputs 3D rotation matrices or elements of $SO(3)$, with the regularization loss to enforce the canonicalization prior leads to remarkably effective results. This approach is demonstrated in our model for rotation-robust point cloud classification and part segmentation, leveraging pretrained PointNet [3] and DGCNN [41] architectures (see Section 4.2).

## 4 Experiments

In this section, we present experimental results on images and point clouds to evaluate our method of achieving equivariance with minimal modifications to pretrained networks. The key benefit of our approach is showcased by demonstrating its robustness when evaluating out-of-distribution data that arise from known transformations applied to the test set.

Our code is available at `https://github.com/arnab39/EquivariantAdaptation`

### 4.1 Image domain

**Experiment Setup.** We use ResNet50 [32] and ViT-Base [5], which are pretrained on ImageNet-1K dataset [43] for our image experiments. These are widely used for image classification tasks, and their pretrained checkpoints are publicly available [1] [2]. We finetune these models on several benchmark natural image classification datasets, including CIFAR10 [31], CIFAR100 [31], and STL10 [42]. Moreover, we incorporate four different strategies to finetune the pretrained models, namely: 1) Vanilla, 2) Rotation Augmentation, 3) Learn Canonicalization, and 4) Prior-Regularized Learned Canonicalization. For the canonicalization function, we use a $C8$-equivariant convolutional network. The details of the architecture are available in Appendix B. We jointly train the canonicalization function and fine-tune the pretrained image classification networks. Our total loss is given by $\mathcal{L}_{total} = \mathcal{L}_{fine-tune} + \beta \cdot \mathcal{L}_{prior}$, where $\mathcal{L}_{prior}$ is defined in Eq. 3, $\mathcal{L}_{fine-tune}$ refers to the cross-entropy loss for classification, and $\beta$ is a hyperparameter which is set to 100.

The *Vanilla* model refers to fine-tuning the pretrained checkpoints using data augmentations such as horizontal flips and small angle rotations, while the *Rotation Augmentation* model covers angles ranging from $-180$ to $+180$ degrees. Although *Rotation Augmentation* model is not equivariant,

---

[1] https://pytorch.org/vision/stable/models/generated/torchvision.models.resnet50.html
[2] https://pytorch.org/vision/stable/models/generated/torchvision.models.vit_b_16.html

our goal was to establish an useful baseline to measure the gain in generalization of our proposed model to the out-of-distribution test dataset resulting from rotating the images. Further, we also report results on a discretized version of *Rotation Augmentation*, mentioned as *C8-Aug.* in Table 2, where the fine-tuning dataset is augmented with the application of group elements in $C8$. We employ the *Learned Canonicalization* method from [30], where the canonical orientation is learned with the task signal only, which in our experiment is the classification. Finally, our proposed *Prior-Regularized Learned Canonicalization* approach adds a prior regularization loss that tries to map all images in the original training dataset to the identity group element $e$. We refer to the final two equivariant techniques as LC and Prior-Regularized LC.

**Evaluation Protocol.** In order to test the robustness of the models on rotations, we introduce $\mathcal{G}$-averaged test set that refers to an expanded dataset obtained by applying all group elements to each image, resulting in a test dataset of size $|\mathcal{G}|$ times the original test set. In this section, we consider $\mathcal{G}$ as $C8$ (cyclic group with 8 rotations), thus $|\mathcal{G}| = 8$. We report the top-1 classification accuracy achieved on the original as well as this augmented test set (referred to as $C8$-Average Accuracy) to evaluate both in distribution and out-of-distribution performance of the models.

**Results.** We report the results of finetuning ResNet50 and ViT on CIFAR10, CIFAR100, and STL10 with various strategies in Table 2. As anticipated, we found that large pretrained networks for images are not robust to rotation transformations, as indicated by the large drop in performance from the accuracy to its $C8$-averaged counterpart for both ResNet50 and ViT. Nevertheless, we observe that ViT is more robust to rotations compared to ResNet50, which has also been observed by [44]. We notice that augmenting with a full range of rotation angles during training improves the $C8$-Average Accuracy as demonstrated by our *Rotation Augmentation* baseline. However, it hurts the accuracy of the prediction network in the original test set and does not guarantee equivariance. Augmenting with necessary rotations in *C8-Augmentation* does not ensure equivariance to $C8$ but retains performance on the original test set and reduces the gap between original and C8-averaged accuracies.

LC guarantees equivariance, which can be seen from the minor difference between the accuracies of the original and augmented test sets. Nevertheless, in every dataset, we can observe a significant drop in accuracy for the original test set. We extensively discussed this issue in Section 3.1. However, with *our Prior-Regularized LC* method, we are able to reduce the gap between the accuracy on the original test set while still being equivariant to rotations. This demonstrates that this prior regularization on LC is a promising direction to improve the performance of large-pretrained models while guaranteeing robustness to out-of-distribution samples resulting from transformations like rotation.

Ideally, the accuracy of the original test set should be nearly identical for both the Vanilla setup and our Prior-Regularized LC method. However, we observed a slight difference between their corresponding accuracies. This disparity arises from the fact that the canonicalization model is unable to map all data points (images) perfectly to the identity element $e$, supported by our observations that the regularization loss for prior matching does not diminish to zero. We hypothesize that this stems from the intentional limitation in the expressivity of the canonicalization function, which is done on purpose in [30] to avoid adding excessive computational overhead to the overall architecture. Finally, we note that due to rotation artifacts, a small difference between $C8$-Average Accuracy and Accuracy on original test set is unavoidable.

### 4.1.1 Instance Segmentation

**Experiment Setup.** We use MaskRCNN [45] and Segment Anything Model (SAM) [21], which are pretrained on Microsoft COCO [22] and SA-1B [21] datasets respectively. MaskRCNN is widely used for instance segmentation task, while SAM is a recently proposed foundational model that can leverage prompts (bounding boxes and key points) for instance segmentation, and their pretrained checkpoints are publicly available [3] [4]. In our experiments, we evaluate these models on COCO 2017 dataset, i.e., report zero-shot performance on validation (val) set and use ground truth bounding boxes as prompts for SAM. For the canonicalization function, we use a $C4$-equivariant WideResNet architecture. The details of the architecture are available in Appendix B. As MaskRCNN and SAM are already trained for the instance segmentation task, we only train the canonicalization function using the prior loss $\mathcal{L}_{prior}$ in Eq. 3 to make them equivariant to $C4$ group.

---

[3]https://pytorch.org/vision/main/models/generated/torchvision.models.detection.maskrcnn_resnet50_fpn_v2.html
[4]https://github.com/facebookresearch/segment-anything

**Evaluation Protocol.** To test the generalization capabilities of these models, we introduce and utilize the $C4$-averaged val set ($C4$ refers to cyclic group with 4 rotations for reducing the complexities with transforming ground truth boxes and masks) along with the original val set. We report the mask-mAP score for mask prediction on both these datasets, denoted respectively as mAP and $C4$-Avg mAP in Table 3.

**Results.** Owing to its pre-training on larger datasets and the use of bounding boxes as prompts, SAM outperforms MaskRCNN in both mAP and $C4$-Avg mAP scores. The difference between mAP and $C4$-Avg mAP scores demonstrates that SAM is more robust than MaskRCNN in the case of these transformations.

However, we observe that there is a difference between these two reported numbers for both models. With our prior regularized LC framework, we can achieve equivariance with any large pretrained model to the desired group (here $C4$) while retaining the performance on the original val set. Further, we analyze the relationship between the expressivity of the canonicalization function and the downstream effect on mAP values on the original val set in Table 3. We compare a $C4$-equivariant convolutional network (G-CNN) with a $C4$-equivariant WideResNet (G-WRN) architecture. We observe that although equivariance is guaranteed, a less expressive canonicalization function leads to decreased performance in the original val set due to its inability to map complex images to identity.

Table 3: Zero-shot performance comparison of large pretrained segmentation models with and without trained canonicalization functions on COCO 2017 dataset [22]. Along with the number of parameters in *canonicalization* and *prediction network*, we report mAP and $C4$-averaged mAP values. † indicates G-CNN and ‡ indicates a more expressive G-WRN for canonicalization.

| Pretrained Large Segmentation Network → | | MaskRCNN (46.4 M) | | SAM (641 M) | |
|---|---|---|---|---|---|
| Datasets ↓ | Model | mAP | $C4$-Avg mAP | mAP | $C4$-Avg mAP |
| COCO [22] | Zero-shot (0 M) | 45.57 | 27.67 | 62.34 | 58.78 |
| | Prior-Regularized LC† (0.2 M) | 35.77 | 35.77 | 59.28 | 59.28 |
| | Prior-Regularized LC‡ (1.9 M) | 44.51 | 44.50 | 62.13 | 62.13 |

## 4.2 Point Cloud Domain

Table 4: Classification accuracy of different pointcloud models on the ModelNet40 dataset [46] in different train/test scenarios and ShapeNet [47] Part segmentation mean IoUs over 16 categories in different train/test scenarios. $x/y$ here stands for training with $x$ augmentation and testing with $y$ augmentation. $z$ here stands for aligned data augmented by random rotations around the vertical/$z$ axis and SO(3) indicates data augmented by random 3D rotations.

| Task → | | Classification | | | Part Segmentation | |
|---|---|---|---|---|---|---|
| Dataset → | | ModelNet40 | | | ShapeNet | |
| Method ↓ | | $z/z$ | $z/$SO(3) | SO(3)/SO(3) | $z/$SO(3) | SO(3)/SO(3) |
| PointNet [3] | | 85.9 | 19.6 | 74.7 | 38.0 | 62.3 |
| DGCNN [41] | | **90.3** | 33.8 | 88.6 | 49.3 | 78.6 |
| VN-PointNet | | 77.5 | 77.5 | 77.2 | 72.4 | 72.8 |
| VN-DGCNN | | 89.5 | 89.5 | 90.2 | 81.4 | 81.4 |
| LC-PointNet | | $79.9 \pm 1.3$ | $79.6 \pm 1.3$ | $79.7 \pm 1.3$ | $73.5 \pm 0.8$ | $73.6 \pm 1.1$ |
| LC-DGCNN | | $88.7 \pm 1.8$ | $88.8 \pm 1.9$ | $90.0 \pm 1.1$ | $78.4 \pm 1.0$ | $78.5 \pm 0.9$ |
| **Ours** (with pretrained PointNet and DGCNN for each task) | | | | | | |
| | | no-aug/$z$ | no-aug/SO(3) | | no-aug/SO(3) | |
| PRLC-PointNet [30] | | $84.1 \pm 1.1$ | $84.3 \pm 1.2$ | | $82.6 \pm 1.3$ | |
| PRLC-DGCNN [30] | | $\mathbf{90.2 \pm 1.4}$ | $\mathbf{90.2 \pm 1.3}$ | | $\mathbf{84.3 \pm 0.8}$ | |

**Datasets.** For our experiments involving point clouds, we utilized the ModelNet40 [46] and ShapeNet [47] datasets. The ModelNet40 dataset comprises 40 classes of 3D models, with a total of 12,311 models. Among these, 9,843 models were allocated for training, while the remaining models were reserved for testing in the classification task. In the case of part segmentation, we employed the ShapeNet-part subset, which encompasses 16 object categories and over 30,000 models. We only train the canonicalization function using the prior loss $\mathcal{L}_{prior}$ in Eq. 7.

**Evaluation protocol.** To ensure consistency and facilitate comparisons, we followed the established conventions set by [49] and adopted by [10] for the train/test rotation setup in the classification and segmentation tasks. The notation $x/y$ indicates that transformation $x$ is applied during training, while transformation $y$ is applied during testing. Typically, three settings are employed: $z/z$, $z/SO(3)$, and $SO(3)/SO(3)$. Here, $z$ denotes data augmentation with rotations around the z-axis during training, while $SO(3)$ represents arbitrary rotations. However, since we regularize the output of the canonicalization with the identity transformation, we trained our canonicalization function and fine-tuned our pretrained model without any rotation augmentation. During inference, we tested on both $z$ and $SO(3)$ augmented test datasets.

**Results.** We present our results on Table 4. Notably, our method showcased superior performance in terms of robustness, outperforming existing methods for point cloud tasks. Specifically, the inclusion of the prior loss has led to a significant improvement in PointNet's performance compared to DGCNN. This observation aligns with our analysis in Section 3.1, where we highlight that training the prediction network with large rotations can hinder its performance and serve as a bottleneck for equivariance within the learnt canonicalization framework. The empirical evidence, particularly in the $SO(3)/SO(3)$ results of vanilla PointNet and DGCNN, where we notice a more pronounced drop PointNet's performance, supports this and strengthens our findings.

## Discussion

Out-of-distribution generalization is an important weakness of state-of-the-art deep models, where an important source of shift in distribution is due to application-specific transformations of the data – from the change of colour palette, magnification and rotation in images to change of volume or pitch in sound. A mechanism for making pretrained models robust to such changes can have a significant impact on their adaptation to different domains. This paper takes the initial steps toward this goal by removing barriers to making the model equivariant through a lightweight canonicalization process. The proposed solution ensures that the pretrained model receives in-distribution data, while canonicalization ensures equivariance and, therefore, adaptability to a family of out-of-distribution samples. Our extensive experimental results using different pretrained models, datasets, and modalities gives insight into this subtle issue and demonstrate the viability of our proposed solution.

## Limitations and Future Work

An important limitation of our approach is the dependency on an equivariant canonicalization network, which imposes restrictions on the range of transformations to which we can adapt. Using the optimization approach of [30] offers more flexibility as it replaces the equivariant network with the outcome of an optimization process. However, this approach can raise efficiency concerns which merit further investigation. Another limitation of the current exposition is the limited group of transformations explored in experiments. In future, we hope to add a comprehensive evaluation of this approach to other groups of transformation. We also aim to address optimization challenges related to prior regularization for $E(2)$-Steerable Networks [15]. By doing so, we aspire to incorporate continuous rotations into our image framework, thereby expanding its capabilities. Moreover, this observation emphasizes the necessity to delve into effect of expressivity of the canonicalization function, as it plays a crucial role in determining the overall performance of our model.

Another promising future direction we hope to explore is more flexible adaptation through canonicalization using examples from a target domain. Such examples from a target domain can, in principle, replace the prior knowledge of the transformation group assumed in equivariant networks, thereby bridging the gap between equivariant deep learning and domain adaptation.

## Acknowledgements

This project was in part supported by CIFAR AI Chairs and NSERC Discovery grant. Computational resources are provided by Mila, ServiceNow Research and Digital Research Alliance (Compute Canada). The authors would like to thank Sebastien Paquet for his valuable feedback.

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

# A Proof of Proposition 1

*Proof.* The cross-entropy is given by

$$\mathbb{E}_{\mathbf{R}\sim q}\left[\log p\left(\mathbf{R}\mid\hat{\mathbf{R}}_p, s_p\right)\right] = \int_{SO(n)} q\left(\mathbf{R}\mid\hat{\mathbf{R}}_q, s_q\right)\log p\left(\mathbf{R}\mid\hat{\mathbf{R}}_p, s_p\right)\mathrm{d}\mathbf{R}, \qquad (8)$$

where $\mathrm{d}\mathbf{R}$ is the invariant Haar measure on $SO(n)$. Here we assume that it is scaled such that $\int_{SO(n)}\mathrm{d}\mathbf{R} = 1$.

We obtain

$$\mathbb{E}_{\mathbf{R}\sim q}\left[\log p\left(\mathbf{R}\mid\hat{\mathbf{R}}_p, s_p\right)\right] = \int_{SO(n)} q\left(\mathbf{R}\mid\hat{\mathbf{R}}_q, s_q\right)\left(s_p\,\mathrm{Tr}\left[\hat{\mathbf{R}}_p^T\mathbf{R}\right] - \log c\left(s_p\right)\right)\mathrm{d}\mathbf{R}, \quad (9)$$

$$\mathbb{E}_{\mathbf{R}\sim q}\left[\log p\left(\mathbf{R}\mid\hat{\mathbf{R}}_p, s_p\right)\right] = s_p\,\mathrm{Tr}\left(\hat{\mathbf{R}}_p^T\,\mathbb{E}_{\mathbf{R}\sim q}\left[\mathbf{R}\right]\right) - \log c\left(s_p\right). \qquad (10)$$

From Theorem 2.2 and Lemma 2.2 of [39], we have

$$\mathbb{E}_{\mathbf{R}\sim q}\left[\mathbf{R}\right] = \frac{d\log c\left(s_q\right)}{ds_q}\hat{\mathbf{R}}_q. \qquad (11)$$

Therefore, we find

$$\mathbb{E}_{\mathbf{R}\sim q}\left[\log p\left(\mathbf{R}\mid\hat{\mathbf{R}}_p, s_p\right)\right] = \frac{d\log c\left(s_q\right)}{ds_q}s_p\hat{\mathbf{R}}_q - \log c\left(s_p\right), \qquad (12)$$

which completes the proof. $\qquad\qquad\square$

# B Architecture Details

We extensively use `escnn` library [15, 50] to design equivariant convolutional architectures.

Each layer of $C8$-**equivariant convolutional network** consists of convolution with regular representation except the first layer, which maps the trivial representation of the $C8$ group to its regular representation. We use equivariant implementation of batch normalization, ReLU activation function, and dropout as proposed in [15, 50]. Major hyperparameters tuned include the number of layers, kernel sizes, dropout, and learning rates. $C4$-equivariant convolutional network utilized in Section 4.1.1 has an identical architecture, with each layer being equivariant to $C4$ instead of $C8$.

$C4$-**equivariant WideResNet** includes repetitive stacking of equivariant versions of *basic* residual blocks on several consecutive *bottleneck* residual blocks (for details on these residual blocks, we refer readers to Figure 1 in [51]). The rest of the architecture details and hyperparameters are identical to the design of $C8$-equivariant convolutional network.

# C Additional analysis experiments

**Augmentation effects** We provide the canonical orientation predicted by Learned Canonicalization (LC) [30] for Rotated MNIST [52] at the start of the training and after training along with original images. The progression of canonized images at the start and after training demonstrates the diminishing effect of augmentation effects when trained with learned canonicalization.

**Effect of prior regularization.** We present a comprehensive analysis of the output distribution over 8 discrete angles predicted by the canonicalization function, both before and after training, on the test set. These findings are depicted in Figure 4 for Learned Canonicalization (LC) [30], and Figure 5 for Prior-Regularized LC. Here, the numbers 0 to 7 correspond to angles that are multiples of 45 degrees, ranging from 0 to 315 degrees, respectively.

We demonstrate that incorporating prior regularization into the canonicalization function aids in mapping the images to the identity prior (represented by 0 angle). This improvement positively impacts the accuracy on the original test set, as evidenced by the results in Table 2. Conversely,

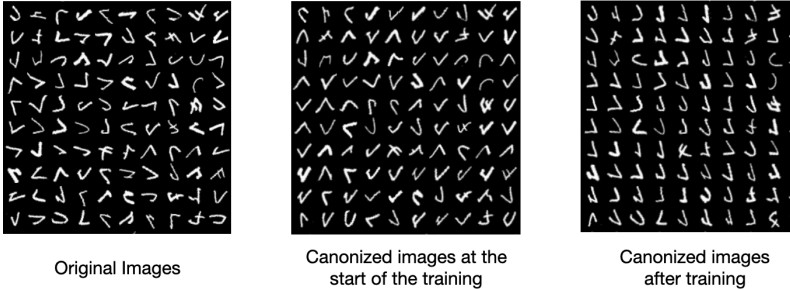

| Original Images | Canonized images at the start of the training | Canonized images after training |

Figure 3: Visualization of the diminishing augmentation effect introduced by learning canonicalization [30] during training for rotated MNIST dataset. In this visualization, the leftmost image represents the original training images. Moving towards the center, we present the canonized images at the beginning of the training process. Finally, the rightmost image unveils the transformation of the canonized images after training the model for 100 epochs.

relying solely on the classification task loss yields no significant alteration in the angle distribution, as the post-training test set distributions remain random.

Additionally, we provide valuable insights into the fraction of images mapped to the identity element in Table 5. It is important to note that the expressivity of the canonicalization function, specifically employing lightweight equivariant networks, contributes to the inability to map all images to the identity elements. This observation calls for further exploration in understanding the role of expressivity and generalization within canonicalization networks with known prior orientations.

Table 5: Fraction of images mapped to identity.

|  |  | Training Completed | |
| Dataset ↓ | Model | ✗ | ✓ |
| CIFAR10 [31] | Learned Canonicalization (LC) [30] | 0.11 | 0.23 |
|  | Prior-Regularized LC (ours) | 0.11 | 0.76 |

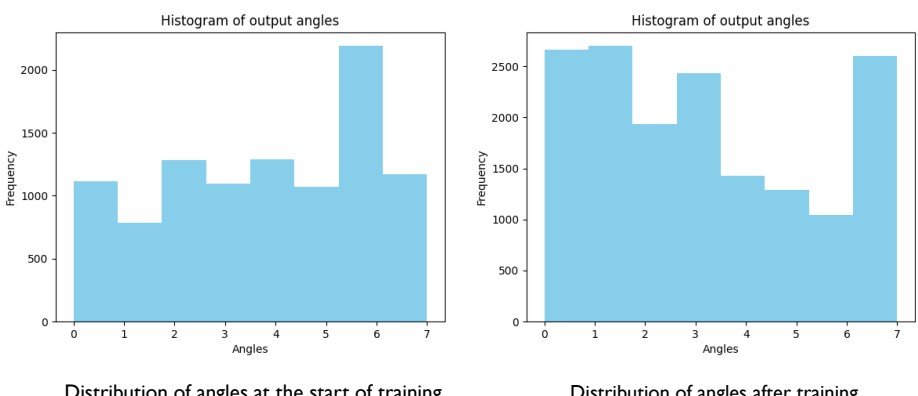

| Distribution of angles at the start of training | Distribution of angles after training |

Figure 4: Distribution of angles output from canonicalization function in $C8$ for Learned Canonicalization [30] for CIFAR10 [31] before and after training. We use indices on the $x$-axis instead of angle values to represent the corresponding multiple of $45°$. Frequency denotes the number of images mapped to a particular multiple of $45°$.

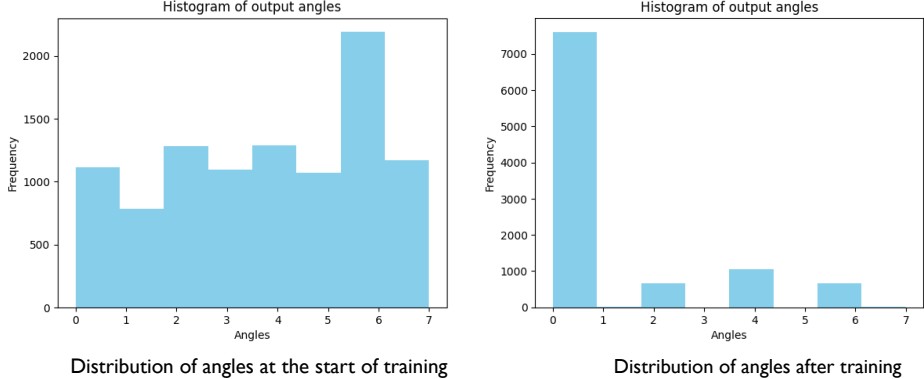

Distribution of angles at the start of training      Distribution of angles after training

Figure 5: Distribution of angles output from canonicalization function in $C8$ for Learned Canonicalization [30] with prior regularization for CIFAR10 [31] before and after training. We use indices on the $x$-axis instead of angle values to represent the corresponding multiple of $45°$. Frequency denotes the number of images mapped to a particular multiple of $45°$.

## D    Optimization challenges on continuous 2D rotations for Images

We investigate the prior introduced in eq. (7) specifically for images. To achieve this, we employ an E-2 steerable network [34] to devise a canonicalization function that generates rotation matrices. Our approach involves initially generating two vector fields, which are subsequently averaged across the spatial dimension to obtain two vectors. By performing Gram-Schmidt orthonormalization on these vectors, we derive a rotation matrix.

In Figure 6, we present the distributions of predicted angles ranging from $-180$ to $+180$. Through our analysis, we observe that the mean and standard deviation of the predicted angles on the test set are $-0.54$ and $80.23$, respectively. This observation signifies that while the prior guides the canonicalization function to output angles close to $0$, there exist instances where angles significantly deviate from this central value. This phenomenon warrants further investigation and a complete understanding of this left as future work.

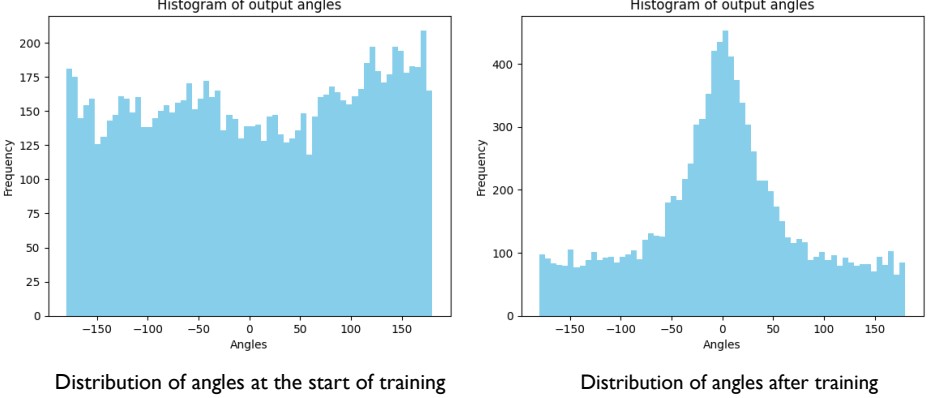

Distribution of angles at the start of training      Distribution of angles after training

Figure 6: Distribution of angles output from steerable canonicalization function in $SO2$ for Learned Canonicalization [30] with prior regularization (eq. (7)) for CIFAR10 [31] before and after training. $x$-axis denotes angles from $-180$ to $+180$. Frequency denotes the number of images mapped to a particular angle.

