# OpenReview forum: "Equivariant Adaptation of Large Pretrained Models"
_NeurIPS.cc/2023/Conference — NeurIPS 2023 poster_

### Official Review · Reviewer_nPdb · 2023-06-22

**Soundness:** 4 excellent
**Presentation:** 3 good
**Contribution:** 3 good
**Rating:** 7
**Confidence:** 4

**Summary:**

The authors tackle an important problem: equivariant NNs are very useful because of capturing important symmetries of the data that lead to higher sample efficiency and more accurate and robust predictions. Yet, EqNNs are very hard to scale because of their specialized architectures. In particular, they cannot benefit from large scale pre-traininig.

The authors propose to use the canonical localization network (an EqNN) from [11] to equip pre-trained model with equivariance. The authors train the localization NN jointly with the prediction network. They test both on the original and augmented test set (with various transformations).

The authors identify that directly applying the approach from [11] does not work because the localization network can transform the input into atypical views for the training dataset. Then they propose a KL regularization term that encourages a delta function at the identify transformation. Their approach works effectively both for discrete and continuous groups.

**Strengths:**

S1: The idea of equipping pre-trained models with equivariance is very timely and potentially useful for large-scale applications.

S2: The regularization term is well-motivated and effective.

S3: The experiments are thorough, including both discrete and continuous groups.

**Weaknesses:**

W1: It would be useful to study your approach in more realistic setting where you do not have to augment the test dataset but rather take an existing test dataset. I'd recommend standard out-of-distribution evaluation on IN-{adversarial, sketch, renditions}, INv2 etc. Another domain for experimentation is in Transfer Learning from IN-1K pertaining. There are some standard datasets, such as Flowers-102, Food, etc.

W2: Since you want to integrate equiavriance with pre-trained models, I think you should also consider SSL pre-trained checkpoints, since they are the foundation of SOTA pre-training (Pre-training on IN-1K may not give you the best features for downstream tasks). Some pre-trained checkpoints are available on github, e.g. https://github.com/rdangovs/essl/tree/main/imagenet/simclr.

W3: As you acknowledged, your model is constrained by the EqNN architectures for the canonical localization network. I wonder if you could consider in your main text discussion more flexible (albeit non-exact) approaches for encouraging symmetry in predictions, such as MSE (https://arxiv.org/abs/2303.02484) and E-SSL (Equivariant Contrastive Learning).

**Questions:**

See the above Weaknesses. My comments should be treated as questions that could improve the quality of your work. I think you have an awesome paper, but I also think that addressing W1-3 could make your work even more impactful.

**Limitations:**

The authors adequately addressed the limitations of their work.

---

> ### Author Rebuttal · Authors · 2023-08-09
>
> We thank the reviewer for their positive assessment of our work and constructive feedback to improve the quality of the paper.
>
> > More realistic settings without augmented test sets:  I'd recommend standard out-of-distribution evaluation on IN-{adversarial, sketch, renditions}, INv2 etc. Another domain for experimentation is in Transfer Learning from IN-1K pertaining. There are some standard datasets, such as Flowers-102, Food, etc.
>
> We would like to highlight that the out-of-distribution in the submission refers to the robustness to transformations that form a group, like rotation of the original images. OOD generalization with other suggestions such as IN-{adversarial, sketch, renditions}, INv2 datasets involve potentially non-linear transformations for which we do not have the corresponding equivariant network required for canonicalization. Extending our framework to transformations beyond groups with approximate equivariance will be an interesting future direction of research.
>
> > Experiments with SSL pretrained checkpoints: [...] I think you should also consider SSL pre-trained checkpoints [...]
>
> We now provide the full fine-tuning results for the E-SSL checkpoint (with best acc1 on the GitHub link shared by the reviewer) on CIFAR10 in Table 3 in the rebuttal pdf. Interestingly, while the Accuracy was similar to vanilla classification checkpoints, the C8-Average Accuracy for E-SSL was poorer. Using our method, we are able to observe similar Accuracy as well as equivariance.
>
> > Non-equivariant architecture for canonicalization: I wonder if you could consider in your main text discussion more flexible (albeit non-exact) approaches for encouraging symmetry in predictions, such as MSE [...] and E-SSL
>
> In addition to the E-SSL results above, we now also provide results for training non-equivariant canonicalization function $c$ in Table 2 in the rebuttal pdf (row titled ConvNet Canonicalizer). We use a simple ConvNet as $c$ and train it to predict the angle for a rotated image, i.e., the prior loss is replaced with an MSE loss between the predicted and actual angle of rotation. As a result, we observe “approximate”  equivariance with a slight drop in the Accuracy over the original test sets.

---

> > ### Comment · Reviewer_nPdb · 2023-08-19
> > **Thanks for the clarifications.**
> >
> > I read the whole rebuttal and would like to maintain my score.
> >
> > A revised paper that incorporates the discussions from the rebuttal would be suitable for publication.
> >
> > Small comment: It would be nice to see OOD tasks in future work because they are more realistic and the extension beyond groups is a meaningful one.

---

### Official Review · Reviewer_vNov · 2023-07-06

**Soundness:** 3 good
**Presentation:** 3 good
**Contribution:** 3 good
**Rating:** 6
**Confidence:** 4

**Summary:**

The paper proposed a fine-tuning based adaptation method for equivariance in pre-trained models. The core of the method is the canonicalization function, which is trained to normalize the input to a common orientation. The paper investigates challenges associated with training canonicalization functions and proposes a canonicalization prior regularization to alleviate those challenges. Experiments in image and point-cloud domains demonstrate that fine-tuning a backbone network with the proposed method improves equivariant properties of a backbone model.

**Strengths:**

1. The paper is very clearly written and easy to follow. All parts of the paper/method are nicely motivated.

2. The paper tackles an important problem of efficient equivariance in neural networks. This problem is especially relevant with regard to large models.

3. The paper provides valuable insights into a practical application of canonicalization functions to achieve equivariance/invariance.


**Weaknesses:**

In no particular order.

1. The paper argues about the equivariant adaptation, but only invariance (and only to the rotation group) is demonstrated in the experiments.

2. Although there is a conceptual novelty in the paper (in the part of analyzing the challenge of applying canonicalization functions), the proposed adaptation approach relies heavily on the prior work of [11]. With this, the methodological novelty of the proposed approach is rather limited.

3. The proposed method requires fine-tuning the whole model together with the canonicalization function. This can be troublesome, especially for large models. With this, I am also wondering if the "equivariant adaptation" is the right name, because the essence of the method seems to be more in equivariant fine-tuning. It would be nice to have experiments, where the backbone model is frozen, and only the canonicalization function is trained.

4. The paper claims out-of-distribution robustness as a key benefit of the method (L224). However, it is not clear how the canonicalization function is itself robust to the distribution shift. If the canonicalization function is trained on cifar10 and tested on cifar100, for example, will it still be able to deliver reasonable distribution over a group orbit? In that sense, it is important to note that the proposed method does not provide guaranteed equivariance as equivariant networks do.

5. I am not sure that comparing methods based on raw Accuracy is a fair evaluation protocol. If the goal of equivariant adaptation is to robustify a model (to a symmetry group of interest), then it seems more appropriate to keep track of the relative change of accuracy from original to G-averaged test sets. Otherwise, it is hard to disentangle if the performance improvement is due to better equivariance or due to improved data augmentation (and a higher accuracy on the non-transformed dataset as a result). This seems to be an important part of the comparison, which is missing from the paper. For example, consider Table 2 results for VIT on Cifar10, the LC model loses 0.2% of its Accuracy, while Regularized LC loses 1% of its Accuracy. Which one would we call more robust?

Others:
1. Possibly useful related work to resolve the issue discussed in L208 - 219. Moskalev et al. LieGG: Studying Learned Lie Group Generators. NeurIPS22 https://arxiv.org/abs/2210.04345
2. Missed related work. Tai et al. Equivariant Transformer Networks. ICML19. https://arxiv.org/abs/1901.11399

UPD: Authors addressed most of my concerns in the rebuttal.

**Questions:**

I suggest authors address weaknesses for the rebuttal.

**Limitations:**

Limitation section nicely highlights limitations of the proposed approach.

---

> ### Author Rebuttal · Authors · 2023-08-09
>
> We thank the reviewer for their detailed review and suggestions.
>
> > Experiments on equivariant tasks: The paper argues about the equivariant adaptation, but only invariance (and only to the rotation group) is demonstrated in the experiments.
>
> We appreciate the reviewer's concern. To address this, we provide results of instance segmentation on COCO 2017 [1] (val set) with Segment Anything Model [2] (SAM) and MaskRCNN [3], where the pre-trained models were not fine-tuned. Our training includes only training the canonicalization function $c$ to map images in the train set of COCO 2017 to identity. As presented in Table 1 in the rebuttal pdf, we observe that our equivariant adaptation obtains mask-mAP scores on the original test set identical to the zero-shot performance of the model while being equivariant to the instance segmentation task. The original models are far from being equivariant, as highlighted by our results. Finally, extending our framework to transformations beyond rotations with approximate equivariance will be an interesting future direction.
>
> > Experiments with frozen backbone:  fine-tuning the whole model together with the canonicalization function. This can be troublesome, especially for large models. [...] nice to have experiments, where the backbone model is frozen, and only the canonicalization function is trained.
>
> We now provide results for instance segmentation in Table 1 in the rebuttal pdf, where the setup is identical to the reviewer’s suggestion, i.e., we freeze the Segment Anything Model and MaskRCNN and only trained the canonicalization function $c$.
>
> > Out of distribution generalization of canonicalization function, and equivariance:  [...] If the canonicalization function is trained on cifar10 and tested on cifar100, for example, will it still be able to deliver reasonable distribution over a group orbit? In that sense, it is important to note that the proposed method does not provide guaranteed equivariance as equivariant networks do [...]
>
> We would like to highlight the fact that the out-of-distribution in the submission refers to the robustness to the rotation of the original images. Since the canonicalization function $c$ is equivariant to rotation groups, contrary to the statement in the review, it is guaranteed that $c$ and the entire pipeline is equivariant and robust to distribution shift. In the particular case of CIFAR 10 and CIFAR 100, since the datasets are identical, their canonicalization networks can be identical and can transfer. In general, out-of-distribution generalization to transformations beyond those handled through equivariance remains an interesting direction for future.
>
> > Fair comparison metrics: I am not sure that comparing methods based on raw Accuracy is a fair evaluation protocol. [...]  it seems more appropriate to keep track of the relative change of accuracy from original to G-averaged test sets.[...]
>
> Both the LC [3] and our Prior-Regularized LC are equivariant, which means, in principle, there should be no gap between accuracy and G-averaged accuracy. Therefore the suggested metric will not be informative. In practice, we observe small discrepancies between Accuracy and G-Averaged Accuracy, which we attribute to the rotation artifacts which destroy information in the image. We will add a sentence to the paper to clarify this.
>
> > Additional related work:
>
> We thank the reviewer for their constructive feedback and suggestion of these related works, which we will incorporate into our work.
>
> $$$$
> $$$$
> [1] Alexander Kirillov, Eric Mintun, Nikhila Ravi, Hanzi Mao, Chloe Rolland, Laura Gustafson, Tete Xiao, Spencer Whitehead, Alexander C. Berg, Wan-Yen Lo, Piotr Dollár, and Ross Girshick. Segment anything, 2023.
>
> [2] Kaiming He, Georgia Gkioxari, Piotr Dollar, and Ross Girshick. Mask r-cnn. In Proceedings of the IEEE International Conference on Computer Vision (ICCV), Oct 2017.
>
> [3] Sékou-Oumar Kaba, Arnab Kumar Mondal, Yan Zhang, Yoshua Bengio, and Siamak Ravanbakhsh. Equivariance with Learned Canonicalization Functions Proceedings of the 40th International Conference on Machine Learning, PMLR 202:15546-15566, 2023
>
> [4] Moskalev et al. LieGG: Studying Learned Lie Group Generators. NeurIPS22.

---

> > ### Comment · Reviewer_vNov · 2023-08-16
> >
> > I thank authors for the response. The rebuttal addresses my concerns, given the rebuttal experiments, clarifications and related work are added to the main paper. I thus raise my score.

---

> > > ### Author Response · Authors · 2023-08-17
> > > **Response to Official Comment by Reviewer vNov**
> > >
> > > We thank the reviewer for their comment.
> > >
> > > >The rebuttal addresses my concerns, given the rebuttal experiments, clarifications and related work are added to the main paper
> > >
> > > We're pleased that our rebuttal has effectively addressed the reviewer's concerns. We'll add all additional experiments, clarifications, and related work in the main text.
> > >
> > > > I thus raise my score
> > >
> > > Since we haven't observed any score increase, we'd kindly like to inquire with the reviewer if they have already made any changes.
> > >
> > > **We see the changes now**

---

### Official Review · Reviewer_45sc · 2023-07-18

**Soundness:** 2 fair
**Presentation:** 2 fair
**Contribution:** 3 good
**Rating:** 3
**Confidence:** 3

**Summary:**

Paper proposes to enforce equivariance/invariance in pretrained models by using the recently proposed canonicalization functions that transform a given input to its canonical form. This allows one to enforce symmetry during the finetuning stage without the need for retraining the model. Paper uses a prior so that the canonicalization network is biased to output identity transformation for the images in the finetuning dataset (i.e., a canonical orientation is defined as the orientation in the finetuning dataset). Empirically, this is shown to be more robust than without the prior.

**Strengths:**

Imposing invariances (via canonicalization) during finetuning rather than pretraining can be very beneficial as different finetuning datasets can have different symmetries.
Authors clearly motivate the challenges of using canonicalization naively for a pretrained model that do not exist when training from scratch (e.g., Figure 3) and propose a simple regularization to solve it. Experiments show better performance across multiple finetuning datasets and pretrained models.


**Weaknesses:**


W1. Details of the overall loss function is not presented or is assumed to be known to the reader. It is harder to evaluate the impact of the prior without these details.

W2. In Section 3.1, the first drawback of Learned Canonicalization is its undesired augmentation effect during the initial phase of finetuning.
- However, this should exist even for the proposed prior-regularized method because the network still begins with a random initialization.
- To solve this, one probably has to encode the prior of outputting identity within the architecture (for example, by defining c(x):= I + r(x) where I is the identity and r(x) initialized with very small weights).

W3. The assumption that the orientations in the finetuning and pretraining datasets are "similar" (line 153) is not well defined. It may be more reasonable to apply the prior using (a small subset of) the pretraining dataset. This will ensure that the finetuning images match the orientation that the model was pre-trained on.

W4. Experiments only evaluate invariance and not equivariance in its general form. I think it is important to demonstrate the advantage of imposing end-to-end equivariance over a pretrained model (as this is different from imposing equivariance in its intermediate representations).

W5. Paper can be strengthened with experiments on other transformations than rotations (but not strictly required).



**Questions:**

Q1. Section 4.1: why is the rotation augmentation baseline applied over all angles from -180 to 180 (lines 235-236)? I believe the group is known to be C8 for the LC and prior-regularized LC methods, and should be used for the baseline as well.

Q2. Section 4.1: Were rotations part of the data augmentations that were performed on the model during its pretraining on ImageNet?

Q3. Section 4.2: VN/CN/PCN- in Table 3 are not defined; I am not sure what these baselines are. Which dataset were the PointNet and DGCNN models pretrained on? Do all the other methods in Table 3 use pretrained models?

Q4. Line 247 says that proposed approach tries to map all images in the "original training dataset" to identity. Should it say "finetuning dataset" instead?


**Limitations:**

Yes.

---

> ### Author Rebuttal · Authors · 2023-08-09
>
> We thank the reviewer for their useful comments on the paper.
>
> > Missing overall loss function equation: Details of the overall loss function is not presented or is assumed to be known to the reader [...]
>
> We thank the reviewer for pointing out the absence of an overall loss function. The total loss function consists of task loss (which is cross-entropy loss) and prior loss. We will add an expression for total loss as $\mathcal{L} = \mathcal{L_{task}} + \lambda * \mathcal{L_{prior}}$, where $\lambda$ is a hyperparameter. We refer the reviewer to eq. (3), which describes the prior loss, and further to line 171, which demonstrates that it reduces to cross-entropy loss over the number of discrete rotations. We set $\lambda$ as 100 in our experiments.
>
> > Effect of initial undesired augmentation in canonicalization model: [...]
>
> We agree with the reviewer that if the prediction network is not frozen when the network begins to train, the problem of the undesired augmentation effect exists in the case of our proposed prior-regularized canonicalization. However, this undesired effect quickly reduces with training with the regularization, as demonstrated in Figures 4 and 5 in the Supplementary material (Appendix C, Effect of prior regularization paragraph). Encoding the prior in the architecture, as suggested by the reviewer, breaks the equivariance of the canonicalization function, and therefore the entire setup is no longer guaranteed to be equivariant.
>
> > Similar orientations in pre-training and fine-tuning dataset: The assumption that the orientations in the finetuning and pretraining datasets are "similar" (line 153) is not well defined. It may be more reasonable to apply the prior using (a small subset of) the pretraining dataset.
>
> We agree with the suggestion and will add a sentence to point out this possibility. While the proposed suggestion is interesting, in the natural image datasets we considered, the canonical orientations are aligned in the finetuning and pretraining datasets. Hence, we assume that this canonical orientation is close to the identity element (or upright images). However, learning or inferring this prior automatically using the pretraining dataset could be an exciting future research direction of this work.
>
> > Experiments with equivariance tasks:  Experiments only evaluate invariance and not equivariance in its general form.
>
> We appreciate the reviewer's concern that the current set of experiments does not evaluate equivariance and rather focuses primarily on invariance tasks. To address this, we provide the result for instance segmentation on COCO 2017 [1] (val set) with Segment Anything Model  [2] (SAM) and MaskRCNN [3], where the pre-trained models were not fine-tuned. Our training includes only training the canonicalization function to map images in the train set of COCO 2017 to identity. As presented in Table 1 in the rebuttal pdf, we observe that our equivariant adaptation obtains mask-mAP scores on the original test set identical to the zero-shot performance of the model while being equivariant to the instance segmentation task.
>
> > Results for fine-tuning with $C8$ augmentations: [...] the group is known to be C8 for the LC and prior-regularized LC methods, and should be used for the baseline as well [...]
>
> We now provide the results for fine-tuning with C8 augmentations in Table 2 in the rebuttal pdf. Our proposed canonicalization method is better than the suggested baseline across all datasets.
>
> > Pre-training setup:  Were rotations part of the data augmentations that were performed on the model during its pretraining on ImageNet
>
> We want to point out that we did not perform the pretraining of large models but rather utilized the widely available checkpoints, as mentioned in line 229. However, the training setup of ResNet-50 and ViT can be found in [4] and [5], which shows that large rotations were not part of data augmentations which explains the poor G-Average Accuracy.
>
> > Description of abbreviations: Expand VN/CN/PCN in Table 3 and do all the other methods in Table 3 use pre-trained models?
>
> We thank the reviewer for pointing this out. Abbreviations VN and CN are taken from [6] and refer to Vector Neuron and Learned Canonicalization (LC). PCN stands for Prior Regularized LC. In the final version, we will change CN and PCN to LC and Prior Regularized LC to make the abbreviations consistent across all domains.
> For classification, we took checkpoints trained on ModelNet40, and for instance segmentation, we took checkpoints trained on ShapeNet dataset. The other methods in the table are trained from scratch. Note that we weren't aware of any foundation model for the PointCloud domain but still wanted to test our idea and show prior regularized finetuning can improve learned canonicalization [6]. We will add these details on pretrained checkpoints in Section 4.2.
>
> > Typo in the submission: Line 247 says [...] Should it say "finetuning dataset" instead?
>
> Thanks for pointing out the typo.
>
> $$$$
> $$$$
>
> [1] Lin, T. et al. Microsoft coco: Common objects in context. In Computer Vision–ECCV 2014: 13th European Conference, Zurich, Switzerland, September 6-12, 2014, Proceedings, Part V 13, pages 740–755. Springer, 2014.
>
> [2] Kirillov, A. et al. Segment anything, 2023.
>
> [3] He, K. et al. Mask r-cnn. In Proceedings of the IEEE International Conference on Computer Vision (ICCV), Oct 2017.
>
> [4] He, K. et al. Deep residual learning for image recognition. In Proceedings of the IEEE conference on computer vision and pattern recognition, pages 770–778, 2016
>
> [5] Dosovitskiy, A. et al. An image is worth 16x16 words: Transformers for image recognition at scale. In International Conference on Learning Representations, 2021.
>
> [6] Kaba, S. et al. Equivariance with Learned Canonicalization Functions Proceedings of the 40th International Conference on Machine Learning, PMLR 202:15546-15566, 2023

---

### Official Review · Reviewer_qZrQ · 2023-07-24

**Soundness:** 3 good
**Presentation:** 3 good
**Contribution:** 3 good
**Rating:** 5
**Confidence:** 4

**Summary:**

The work proposes a method for making large pre-trained models equivariant to specified group actions. The method uses the canonicalization approach, where the input is oriented to a specific orientation before being fed to the pre-trained model with the help of a trainable canonicalization network. The authors use a prior distribution over the group elements to regularize the output of the canonicalization network. The proposed technique shows robustness with respect to the group actions in different downstream tasks, such as classification and part segmentation.

**Strengths:**

1. The paper pursues a novel approach to make pre-trained models scale equivariant.
2. The proposed technique is simple and effective, making it well-suited for practical adaptation.
3. The work is well-written and easy to follow.

**Weaknesses:**

1. Experimental Setup: One major drawback of this work is the experimental setup. The authors used priors over the discrete group elements (i.e., 8 discrete rotations of C8 for the image domain) for the experiments. During the evaluation on the test set, the images were augmented by the action of group C8. This approach is not appropriate because we are providing the model with information about the test set augmentation during training.

    On the other hand, for the data augmentation setup, the baseline models were fine-tuned using random rotations between -180 to 180 degrees (i.e., all possible rotations). In this case, no information about the test set augmentation was given to the baseline.

    This difference in augmentation approaches makes the comparisons inappropriate. An appropriate comparison should either:

    1. Use a uniform continuous prior over the group elements.
    2. Perform data augmentation only with the discrete group actions while training the baseline.

    Option 1 is more practical because in the real world, we often do not know specific priors over the group actions.

2. Recent studies have shown that fine-tuning large pre-trained models on small datasets hampers downstream tasks [1] and makes the model vulnerable to out-of-distribution data. Experiments with frozen pre-trained backbones and trainable classification/segmentation modules would be more justifiable.

1.Fine-Tuning can Distort Pretrained Features and Underperform Out-of-Distribution

**Questions:**

1. Should the Kronecker delta functions in line 172 be normalized by the number of group elements for discrete priors?
2. What is the specific structure of the **canonicalization** module used in this work?

---

> ### Author Rebuttal · Authors · 2023-08-09
>
> We thank the reviewer for their useful feedback. Below we address the reviewer's concerns.
>
> > Concerns on experimental setup: [...] appropriate comparison should [...] perform data augmentation only with the discrete group actions while training the baseline.
>
> We thank the reviewer for pointing out this issue. We now provide the results for fine-tuning with C8 augmentations in Table 2 in the rebuttal pdf. Our proposed canonicalization method is better than the suggested baseline across all datasets and significantly so in the case of CIFAR100. Regarding uniform continuous prior, Appendix C details some of the optimization challenges when incorporating this with steerable networks. As mentioned in lines 314-316, a complete understanding of this and an extension of our method to continuous rotation in images is left as a future work.
>
> > Fine-tuning and frozen backbone architectures:  [...] experiments with frozen pre-trained backbones and trainable classification/segmentation modules would be more justifiable
>
> We understand the justification of the reviewer’s suggestion to use frozen pre-trained backbones and trainable modules. To address this concern for more realistic settings, we provide the result for instance segmentation on COCO 2017 [1] (val set) with Segment Anything Model [2] (SAM) and MaskRCNN [3], where the pre-trained models were not fine-tuned (due to computational resource constraint). Our training includes only training the canonicalization function to map images in the train set of COCO 2017 to identity. As presented in Table 1 in the rebuttal pdf, we observe that our equivariant adaptation obtains mask-mAP scores on the original test set identical to the zero-shot performance of the model while being equivariant to the instance segmentation task.
>
> > Should the Kronecker delta functions in line 172 be normalized by the number of group elements for discrete priors?
>
> Since we are putting all probability mass of the prior on identity for natural images, we are using a Kronecker delta function. If we use the normalized Kronecker delta function as suggested, the prior will not be a probability distribution.
>
> > What is the specific structure of the canonicalization module used in this work?
>
> We thank the reviewer for pointing out this omission, and we will add the following architectural details to the appendix of the final version of our paper:
>
> We extensively use the $\texttt{escnn}$ library [4, 5] to design equivariant convolutional architectures.
> - For classification experiments, we use a $C8$-equivariant convolutional network; each layer consists of convolution with regular representation except the first layer, which maps the trivial representation of the $C8$ group to its regular representation. We use equivariant implementation of batch normalization, ReLU activation function, and dropout as proposed in [4, 5].
> - For instance segmentation, we use a $C4$-equivariant WideResNet, which includes repetitive stacking of equivariant versions of *basic* residual blocks on several consecutive *bottleneck* residual blocks (for details on these residual blocks, we refer readers to Figure 1 in [6]). The rest of the architecture details and hyperparameters are identical to the design of the $C8$-equivariant convolutional network.
>
> $$$$
> $$$$
>
> [1] Tsung-Yi Lin, Michael Maire, Serge Belongie, James Hays, Pietro Perona, Deva Ramanan, Piotr Dollár, and C Lawrence Zitnick. Microsoft coco: Common objects in context. In Computer Vision–ECCV 2014: 13th European Conference, Zurich, Switzerland, September 6-12, 2014, Proceedings, Part V 13, pages 740–755. Springer, 2014.
>
> [2] Alexander Kirillov, Eric Mintun, Nikhila Ravi, Hanzi Mao, Chloe Rolland, Laura Gustafson, Tete Xiao, Spencer Whitehead, Alexander C. Berg, Wan-Yen Lo, Piotr Dollár, and Ross Girshick. Segment anything, 2023.
>
> [3] Kaiming He, Georgia Gkioxari, Piotr Dollar, and Ross Girshick. Mask r-cnn. In Proceedings of the IEEE International Conference on Computer Vision (ICCV), Oct 2017.
>
> [4] Maurice Weiler and Gabriele Cesa. General e (2)-equivariant steerable cnns. Advances in Neural Information Processing Systems, 32, 2019.
>
> [5] Gabriele Cesa, Leon Lang, and Maurice Weiler. A program to build e(n)-equivariant steerable CNNs. In International Conference on Learning Representations, 2022.
>
> [6] Sergey Zagoruyko and Nikos Komodakis. Wide residual networks. arXiv preprint arXiv:1605.07146, 2016.

---

> > ### Comment · Reviewer_qZrQ · 2023-08-16
> > **Response to the Rebuttal**
> >
> > I thank the authors for the detailed response and additional experiments. It seems finetuning the baselines with C8 augmentation reduced the performance gap between the proposed method and the baseline. And should be added to the main text. I am increasing my score.

---

> > > ### Author Response · Authors · 2023-08-17
> > > **Response to Reviewer qZrQ**
> > >
> > > We thank the reviewer for increasing the score.
> > >
> > > >It seems finetuning the baselines with C8 augmentation reduced the performance gap between the proposed method and the baseline. And should be added to the main text
> > >
> > > We will add the C8 baseline to the main text as suggested. While employing C8 augmentation as a baseline has slightly narrowed the performance gap, our method continues to exhibit consistent performance improvement across all datasets, particularly pronounced in CIFAR 100.
> > >
> > > Given that the reviewer still rated it borderline, we would appreciate any concerns/questions they might have in mind that could enhance their evaluation and potentially lead to an increased score.

---

### Author Rebuttal · Authors · 2023-08-09

We thank all the reviewers for their valuable and constructive feedback. We are pleased to see that they found our work simple (reviewers **45sc**, **qZrQ**),  novel, and effective (reviewers **qZrQ**, **nPdb**)  in addressing a relevant, important (**vNov**) and timely (**nPdb**) problem.  In the general response, attached rebuttal PDF, and individual reviewer responses, we believe to have addressed most if not all, reviewer questions and concerns. In particular, we are including significant new results on the Segment Anything Model [1] (SAM) and MaskRCNN [2] for equivariant segmentation, as well as additional baselines and ablations for invariant classification. These new results address common concerns on 1) equivariant tasks, 2) adapting with completely frozen weights, 3) missing baseline with C8 augmentation, 4) using E-SSL embeddings 5) a new baseline using a non-equivariant canonicalization network.

Below, we give more context on these new experiments and refer the reviewers to the new and updated tables in the attached rebuttal PDF:

1. We provide results of instance segmentation on COCO 2017 [3] (val set) with Segment Anything Model (SAM) and MaskRCNN, where the pre-trained models were not fine-tuned, in part, due to the size of these pre-trained models. Our training includes only training the canonicalization function to map images in the train set of COCO 2017 to identity. As presented in Table 1 in the rebuttal pdf, we observe that our equivariant adaptation obtains mask-mAP scores on the original test set identical to the zero-shot performance of the model while being equivariant to the instance segmentation task. As this task is an equivariant task where we don’t finetune the large foundational prediction model, this addresses reviewers' concerns about only focussing on invariant tasks and fine-tuning the prediction network. We show that we can achieve equivariance by just training the canonicalization function (canonicalizer) and attaching it to a foundation model.

2. Based on reviewers’ feedback, we provide some additional baselines for the classification tasks on the image datasets:
    - we have added a baseline with C8 augmentation during fine-tuning, and the corresponding results are available in Table 2 in the rebuttal pdf (row titled C8-Aug.).
    - CIFAR10 results using the suggested E-SSL [4] checkpoint for ResNet-50 also appear in Table 3 in the rebuttal pdf.
    - Finally, based on the suggestion of Reviewer **nPdb**, we explore a setup for pre-trained ResNet-50 where we replace an equivariant canonicalizer with a non-equivariant convolutional network (ConvNet Canonicalizer). We will add the architectural details in the appendix of the final version of our paper. We train the ConvNet Canonicalizer to predict the angle for a rotated image, i.e., the prior loss is replaced with an MSE loss between the predicted and actual angle of rotation. As a result, we observe “approximate”  equivariance with a slight drop in the Accuracy over the original test sets. The accuracy values are provided in Table 2 (row titled ConvNet Canonicalizer).

$$$$
We plan to include all these new results in the final version of the paper. We hope that the reviewers can reassess our work and its contributions in light of this.

$$$$
$$$$

[1] Alexander Kirillov, Eric Mintun, Nikhila Ravi, Hanzi Mao, Chloe Rolland, Laura Gustafson, Tete Xiao, Spencer Whitehead, Alexander C. Berg, Wan-Yen Lo, Piotr Dollár, and Ross Girshick. Segment anything, 2023.

[2] Kaiming He, Georgia Gkioxari, Piotr Dollar, and Ross Girshick. Mask r-cnn. In Proceedings of the IEEE International Conference on Computer Vision (ICCV), Oct 2017.

[3] Tsung-Yi Lin, Michael Maire, Serge Belongie, James Hays, Pietro Perona, Deva Ramanan, Piotr Dollár, and C Lawrence Zitnick. Microsoft coco: Common objects in context. In Computer Vision–ECCV 2014: 13th European Conference, Zurich, Switzerland, September 6-12, 2014, Proceedings, Part V 13, pages 740–755. Springer, 2014.

[4] Rumen Dangovski, Li Jing, Charlotte Loh, Seungwook Han, Akash Srivastava, Brian Cheung, Pulkit Agrawal, and Marin Soljacic. Equivariant self-supervised learning: Encouraging equivariance in representations. In International Conference on Learning Representations, 2022.

---

### Decision · Program_Chairs · 2023-09-21

**Decision:**

Accept (poster)

**Comment:**

Three out of four reviewers advocate for the acceptance of this paper. The reviewer that recommended rejection did not respond during the rebuttal discussion period. The AC have checked over the author's rebuttal and believe that the reviewer’s concerns are partially addressed. Overall, the AC finds this paper interesting and has practical implications. The AC trusts that the authors will include all the results and polish the writing.